# Use of Carbonated Water as Kneading in Mortars Made with Recycled Aggregates

**DOI:** 10.3390/ma15144876

**Published:** 2022-07-13

**Authors:** David Suescum-Morales, José Ramón Jiménez, José María Fernández-Rodríguez

**Affiliations:** 1Departamento de Ingeniería Rural, Escuela Politécnica Superior de Belmez, Universidad de Córdoba, 14240 Córdoba, Spain; p02sumod@uco.es; 2Departamento de Química Inorgánica e Ingeniería Química, Escuela Politécnica Superior de Belmez, Universidad de Córdoba, 14240 Córdoba, Spain

**Keywords:** carbonated water, CO_2_ sequestration, accelerated carbonation, circular economy, construction demolition waste

## Abstract

The increased concern about climate change is revolutionising the building materials sector, making sustainability and environmental friendliness increasingly important. This study evaluates the feasibility of incorporating recycled masonry aggregate (construction and demolition waste) in porous cement-based materials using carbonated water in mixing followed (or not) by curing in a CO_2_ atmosphere. The use of carbonated water can be very revolutionary in cement-based materials, as it allows hydration and carbonation to occur simultaneously. Calcite and portlandite in the recycled masonry aggregate and act as a buffer for the low-pH carbonated water. Carbonated water produced better mechanical properties and increased accessible water porosity and dry bulk density. The same behaviour was observed with natural aggregates. Carbonated water results in an interlaced shape of carbonate ettringite (needles) and fills the microcracks in the recycled masonry aggregate. Curing in CO_2_ together with the use of carbonated water (concomitantly) is not beneficial. This study provides innovative solutions for a circular economy in the construction sector using carbonated water in mixing (adsorbing CO_2_), which is very revolutionary as it allows carbonation to be applied to in-situ products.

## 1. Introduction

Cement and concrete are used in building and construction work worldwide. The construction industry producing cement and concrete can be regarded as the world’s largest industry [1,2,3]. Construction is responsible for a great amount of CO_2_ release [4,5]. One tonne of cement emits approximately 0.6 to 0.8 tonnes of CO_2_ [1,6,7,8]. The CO2 concentration in the atmosphere has increased from 280 to 420 ppm in 2020 [9]. The incorporation of waste and the proper use of resources are leading global challenges to control the negative environmental impact of cement and concrete and preserve the planet.

High-emission countries are actively exploring carbon capture and utilisation (CCU) or storage technologies. CCU is a novel method to reduce CO_2_ and turn CO_2_ into a commercially interest product [10,11,12]. The carbonation and chemical reactions in cement-based materials (CBMs) (Equations (1) to (5)) [13,14,15,16,17,18] occurs with CO_2_, affects cement hydration products, and increases CaCO_3_ production [5,19,20]. As early as 1970, the idea of CO_2_ capture through carbonation with CBM appeared [21]. Carbonation of CBM, as an alternate to CCU, reduces water absorption and curing time (useful in the precast industry), increases density, and improves fragmentation resistance and mechanical properties [22,23,24].
(1)$$\mathrm{Ca}\;{\left(\mathrm{OH}\right)}_{2}+{\mathrm{CO}}_{2}\to {\mathrm{CaCO}}_{3}+H_{2}O$$
(2)$$C-S-H+{\mathrm{CO}}_{2}\to {\mathrm{CaCO}}_{3}+{\mathrm{SiO}}_{2}\cdot {\mathrm{nH}}_{2}O$$
(3)$$4\mathrm{Cao}\cdot {\mathrm{Al}}_{2}O_{3}\cdot 13H_{2}O+4{\mathrm{CO}}_{2}\to 4{\mathrm{CaCO}}_{3}+2\mathrm{Al}{\left(\mathrm{OH}\right)}_{3}+10H_{2}O$$
(4)$$3\mathrm{CaO}\cdot {\mathrm{SiO}}_{2}+3{\mathrm{CO}}_{2}+{\mathrm{nH}}_{2}O\to {\mathrm{SiO}}_{2}\cdot {\mathrm{nH}}_{2}O+3{\mathrm{CaCO}}_{3}$$
(5)$$2\mathrm{CaO}\cdot {\mathrm{SiO}}_{2}+2{\mathrm{CO}}_{2}+{\mathrm{nH}}_{2}O\to {\mathrm{SiO}}_{2}\cdot {\mathrm{nH}}_{2}O+2{\mathrm{CaCO}}_{3}$$

The concentration of CO_2_ in the environment must be increased for accelerated carbonation in different ways [14,15,18,23,24,25,26,27,28,29,30]. To increase this level of CO_2_, a carbonation chamber is usually necessary [13] with or without pressure [31,32], with different levels of CO_2_ or even submerging the samples in mixtures of different gases [33,34,35]. A review on the effect of CO_2_ in cement-based materials on the physico-mechanical properties was previously described in another study by Suescum-Morales et al. [36]. The carbonation rate is determined by the diffusion of CO_2_ gas in the samples [35]. Carbonation products make samples more dense, preventing the easy entry of CO_2_. The need for a high amount of CO_2_ in the curing environment implies the need for an accelerated carbonation chamber. An attractive alternative is to apply carbonation technology during cement kneading. To avoid the difficulty of CO_2_ diffusion and make CBM carbonation apply to in-situ products, the kneading water is replaced by carbonated water (water with high CO_2_ content). Thus, it is possible to start the carbonation simultaneously with the hydration process of the cement and increase carbonation [20,37]. Furthermore, the hydration reaction of the cement occurs much faster than that under normal curing conditions [5].

Construction and demolition waste (CDW) is produced during the demolition phases of several types of construction building or infrastructures (over 30 billion tonnes per year worldwide) [13,38,39]. CDW is composed of several types of waste, in addition to concrete and ceramics, such as glass, stone, bituminous material, and others. Recycled aggregates (RAs) are obtained from CDW with appropriate treatment (recycling plant treatment). A possible simple classification of RA may be made, in a simple way [25]: (i) if the waste is ceramic, the aggregate might be called recycled ceramic aggregates; (ii) if is concrete waste, may be called recycled concrete aggregates (RCA); and (iii) if it is a mixture of the two above, mixed recycled aggregates (MRA) [36,40].

Mixed recycled aggregate (MRA) is the most widely produced RA in the world. The non-existence of regulations and different sources is still limited the use of MRA [41]. Recycled masonry aggregate (RMA) differs from natural aggregates (NAs) mainly in terms water absorption, higher porosity, and lower density [42,43]. MRA has had different uses: as aggregates for masonry mortar, and as an aggregate for alkaline activated material or CBM [41,42,43,44,45,46,47,48]. RMA is a type of MRA obtained from screening and crushing walls waste [36,49,50,51,52].

There are two ways to produce accelerated carbonation in RCA [14]: in the aggregate itself [7,42,53,54,55] or in the mixture of RCA with Portland cement [24,27,28,56]. However, these studies do not investigate the effect of CO_2_ on CBMs made with RMA or RCA using carbonated water as kneading water. This research would fill this information gap. Nor has any literature been found that studies the simultaneous use of carbonated water and CO_2_ curing.

This study mainly investigates the physico-mechanical properties of a porous CBM with RMA, and carbonated water as kneading water and for subsequent curing in CO_2_. To observe the effect of carbonated water on the microstructure of the hardened samples, with NA and RMA, and cured under both regimes, scanning electron microscopy (SEM), energy dispersive X-ray spectroscopy (EDS), and backscattered electron (BSE) were performed. Thermogravimetric analysis and differential thermal analysis (TGA/DTA) was also performed to determine the amount of CaCO_3_ in all cases. No studies have been found that simultaneously use carbonated water as kneading water, under accelerated curing, and using RMA as aggregate. The production of precast or in-situ non-reinforced CBM products could be possible with the following approach: incorporation of waste (RMA), with the added value to CO_2_, and inclusion of carbonated water as kneading water, which can be very revolutionary.

## 2. Materials and Methods

### 2.1. Materials

NA was used as a reference. NA and RMA were used in previous research [25,36,49,50]. The components of RMA according to UNE EN 933-11:2009 were: red ceramic bricks (53.9%), masonry mortar (39.8%), unbound aggregates (5.7%), concrete (0.4%), and gypsum particles (0.2%). Water absorption and dry bulk density (DBD) were measured according to UNE-EN 1097-6:2013 [57]. DBD for NA and RMA was 2.63 g·cm^−3^ and 2.14 g·cm^−3^, respectively. The amount of CaCO_3_ for NA and RMA (197 and 239 kg/m^3^ respectively) was calculated using TGA/DTA. Water absorption was 0.79% for NA and 9% for RMA. CEM II/A-V 42.5 R was used as cement [58] with a DBD of 2.89 g·cm^−3^ according to the characteristics provided by the manufacturer.

As kneading water, two types of commercial water were used: normal water (H_2_O) and carbonated water (CO_2_·H_2_O), both from the manufacturer Fuente Primavera, Spain. For H_2_O, the pH value was 7.7 and the initial concentration of CO_2_ was between 0.2–0.5 mg·L^−1^. For CO_2_·H_2_O, these values were 4.8 (for pH) and 14.1–14.4 mg CO_2_·L^−1^.

### 2.2. Experimental Design and Curing Conditions

NA and RMA were sieved: 2/4, 1/2, 0.5 /1, 0.25/0.5, 0.125/0.25, and 0/0.125 fractions to reconstitute the lower limit indicated by ASTM C 144-04 [59]. Two gaps were achieved by deleting the following fractions: 0.25/0.5 and <0.125 mm. Its mineral skeleton (with two gaps) can facilitate the input of CO_2_ and total carbonation of the mix (more porous). Table 1 shows the reconstituted particle size distribution of NA.

Equations (6) and (7) calculate the dry mass of each component:(6)$$Dry\;mass\;of\;cement=\frac{V\cdot 1\cdot \rho_{rd\;cement}}{6}$$
(7)$$Dry\;mass\;of\;NA=\frac{V\cdot 5\cdot \rho_{rd\;natural\;aggregate\;}\;}{6}$$
where ρ*_rd cement_* = 2.89 g/cm^3^ and ρ*_rd natural aggregate_*= 2.63 g/cm^3^, which are the DBD of cement and NA, respectively. A volume (V) of 1600 cm^3^ was manufactured in each mix. According to Equation (7), the mass of NA was 3507 g. The mass of each fraction for NA (Table 2) were obtained according to the 2 gap realized in Table 1. A total substitution of NA by RMA in volume fraction by fraction was realized. To replace NA by RMA, the bulk density was used [60].

The samples were subjected to two curing environments (both with 21 ± 2 °C and 65 ± 10% of relative humidity): (i) normal climatic chamber (CC) and (ii) accelerated carbonation chamber (CO_2_·C). For the CC, the CO_2_ concentration was 0.04%, and for the CO_2_·C it was 5%. The CO_2_ used for this condition was provided by Linde (99.99995% purity). Figure 1 shows the experimental design carried out.

### 2.3. Kneading Process

Table 3 shows the composition of the mixes studied. The aggregates were pre-saturated, according to the water absorption of each one of them (NA or RMA). Therefore the *w*/*c* ratio used can be considered as effective. The kneading process was in accordance with previous research [25,36].

Prismatic 40 × 40 × 160 mm casts were used [61]. The samples were keep in the mould for 3 h. The samples were covered to prevent CO_2_ input/output during this time. After this time (3 h), the samples were demoulded because the aim was to demould the samples very quickly, similar to what happens in a precast plant. According to Pan et al. [35], this pre-curing time is crucial to avoid water-saturated capillary pores resulting in a low penetration rate of CO_2_ for the samples cured in CO_2_·C. The samples were then cured in two chambers: CC and CO_2_·C for 1, 3, and 7 days of curing.

### 2.4. Test Methods

X-ray fluorescence spectrometry analysis (XRF) was realized with ZSX PRIMUS IV, Rigaku equipment. A Bruker D8 Discover A 25 diffractometer were used for to obtain X-ray diffraction (XRD) patterns. A CuKα radiation (λ = 1.54050 Ȧ; 40 Kv; 30 mA) was used and scan angles between 10° to 70° (2θ) were programmed. The speed used was of 0.02 2θ min^−1^. For identifying the diffractograms, the International Database ICDD 2003 was used [62].

TGA/DTA were performed using a Setaram Setys Evolution 16/18 instrument with a resolution of 0.002–0.02 µg. The heating increase was 5° min^−1^.

The flexural (FS) and compressive strength (CS) were obtained according to the European Standard EN 1015-11 [61] for 1, 3, and 7 d of curing. The dry bulk density (DBD) of hardened samples was determined according to European Standard EN 1015-10 [63]. Accessible porosity for water (APW) were measured according to European Standard UNE 83980 [64].

The morphology and composition of the mixes with NA and RMA under the CC regime were studied using H_2_O and CO_2_·H_2_O. SEM, EDS, and BSE were obtained using JEOL JSM 7800F at the age of 7 d. The objective was to observe the effect of carbonated water on the microstructure of the hardened samples, with NA and RMA cured under CC regime. They were then sputtered with gold to obtain the maximum image quality.

All the above tests were carried out in triplicate.

## 3. Results and Discussion

### 3.1. Characterization of Raw Materials

Figure 2 shows the XRD patterns of NA, RMA, and cement. Quartz (SiO_2_) (05-0490) [62] was the main phase for NA and RMA. Other minority phases were also found and were described in greater detail in other research [25,36]. The diffractogram of the cement was in agreement with the finding of other authors [65,66,67,68,69]. Table 4 shows the XRF results found for NA, RMA, and cement, which are in agreement with the phases found in XRD.

### 3.2. Compressive and Flexural Strength

Figure 3 shows the CS results for the four mixes and two curing regimes at ages of 1, 3, and 7 days of curing. When comparing NA-H_2_O-CC with NA-CO_2_·H_2_O-CC at 1 day of age, CS decreased by 8.6%. According to Valdemir dos Santos et al. [20], this result is related to the reduced AFm formation in the microstructure during the early hydration period [20]. Similar results were reported by Lippiatt et al. [5] in a cement paste aged 1 d using carbonated water. In addition, the low pH value of carbonated water (4.8) can negatively affect the strength [70], delay the setting [71], and produce changes in the cement paste structure [72]. It is possible that a low pH leads to the reduction of hydrated calcium silicate and hydrated calcium aluminate because the reactions of equations 8–10 occur. Additionally, the amount of Portlandite present decreases. Therefore, the structure of the cement paste will be weaker. Nevertheless, at 3 and 7 d, an increment of 18% and 12.5% was obtained, respectively, when using carbonated water for the NA mixture and CC regime. When kneading cement and water, the pH increases rapidly, and this solution becomes saturated with Ca(OH)_2_ after 24 h [73,74]. Furthermore, the calcite phase found in NA (Figure 2) can act as a buffer when added to carbonated water, as observed by Lippiatt et al. [5] to achieve simultaneous hydration and carbonation in cement. This saturated solution of Ca(OH)_2_, together with CO_2_ in the carbonated water, favoured the carbonation reaction and increased CS at 3 and 7 d [22,23,24]. Equations (8)–(10) show the chemical reaction with carbonated water [20].
(8)$$CO_{2\left(gaseous\right)}+2OH^{-}{}_{\left(gaseous\right)}↔CO_{3}{{}^{2-}}_{\left(aqueous\right)}+H_{2}O_{\left(liquid\right)}$$
(9)$$Ca{\left(OH\right)}_{2\left(aqueous\right)}↔Ca^{2+}{}_{\left(aqueous\right)}+2OH^{-}{}_{\left(aqueous\right)}$$
(10)$$Ca^{2+}{}_{\left(aqueous\right)}+CO_{3}{{}^{2-}}_{\left(aqueous\right)}+H_{2}O_{\left(liquid\right)}\;↔CaCO_{3\left(solid\right)}+H_{2}O_{\left(liquid\right)}$$

Compared to NA, RMA mixture under CC regime, using carbonated and normal water (RMA-H_2_O-CC and RMA-CO_2_·H_2_O-CC), had slightly lower CS. This loss of mechanical properties agreed with other studies when the percentage of substitution of NA for RA was 100% [14,75,76,77,78,79]. However, compared to normal water, the carbonated water was beneficial in this case for all ages of curing (19.3%, 12.1%, and 4.4% for 1, 3, and 7 d, respectively) due to the presence of CaCO_3_ and Ca(OH)_2_ in RMA (Figure 2). These phases act as a buffer of carbonated water [5], increase pH, avoid the loss of mechanical resistance, and delay hydration [70,71] that occur in the mixture with NA with 1 d of curing. Thus, carbonated water with RMA can improve the mechanical strength under the CC regime.

For NA and RMA mixtures, the increase in CS in samples cured with CO_2_·C (NA-H_2_O-CC vs. NA-H_2_O-CO_2_·C and RMA-H_2_O-CC vs. RMA-H_2_O-CO_2_·C) agree with the results in [13,14,18,24,27,28,56,80,81,82]. For NA mixtures, with 1 d and under CO_2_·C, when using carbonated water in the kneading, compared with normal water, decreased the CS by 38.77% (NA-H_2_O-CO_2_·C vs. NA-CO_2_·H_2_O-CO_2_·C). The low pH value of carbonated water along with accelerated carbonation (CO_2_·C) results in a negative effect on strength [70] and delayed setting [71], which lowers the pH values of the mix [83,84,85]. For 3 d of curing, the effect of carbonated water on CS was still negative. For 7 d of curing, an increment of 13.3% was observed. This could indicate the regulation of pH [22,23,24] and the carbonation of the sample.

The same behaviour was observed in the samples with RMA (RMA-H_2_O-CO_2_·C vs. RMA-CO_2_·H_2_O-CO_2_·C), although with minor decreases for 1 and 3 d. This is again due to CaCO_3_ and Ca(OH)_2_ in RMA (Figure 2) acting as a buffer of carbonated water [5], maintaining a pH higher than that with NA. Thus, carbonated water under accelerated carbonation (with NA and RMA) is beneficial only after 7 d of curing.

The FS results for all the mixes under CC and CO_2_-C at the ages of 1, 3, and 7 d, in Figure 4, reveal the same trend as CS.

### 3.3. DBD and APW

DBD and APW are shown in Figure 5 for 7 d of curing under CC and CO_2_·C. On the NA mixture, under the CC regime, the use of carbonated water as kneading water, compared with normal water, incremented the DBD by 3.3%. This result agrees with the increase in the mechanical properties in Figure 3 and Figure 4. Carbonated water favours the carbonation reaction at 7 d of curing, increasing the DBD [22,23,24]. The APW also increased by 5.28% when using carbonated water during kneading. This result is in accordance with Valdemir et al. [20], who found that CO_2_ released by the carbonated water could generate additional porosity. The same behaviour was observed with RMA (RMA-H_2_O-CC and RMA-CO_2_·H_2_O-CC), in which DBD increased by 0.8% and APW by 19.06%.

For the RMA mixture under the CC regime, when using carbonated and normal water (RMA-H_2_O-CC and RMA-CO_2_·H_2_O-CC), DBD and APW were higher than those in the same mixtures with NA. This agrees with the lower particle dry density, and higher water absorption of RMA reported in [51,79,86].

For the NA and RMA mixtures, using normal water, an increase in DBD and a decrease in APW were observed for samples cured in CO_2_·C (NA-H_2_O-CC vs. NA-H_2_O-CO_2_·C and RMA-H_2_O-CC vs. RMA-H_2_O-CO_2_·C). These mechanical properties could be due to sample carbonation, as observed in [13,14,18,24,27,28,56,80]. Carbonated water for kneading water under accelerated carbonation, compared to normal water, increased the DBD and APW (NA-H_2_O-CO_2_·C vs. NA-CO_2_·H_2_O-CO_2_·C and RMA-H_2_O-CO_2_·C vs. RMA-CO_2_·H_2_O-CO_2_·C). These results agree with the mechanical properties observed in Figure 3 and Figure 4.

### 3.4. XRD

XRD obtained for NA using normal and carbonated water as kneading water under CC are shown in Figure 6. For normal water at 1 d, the main phases found were quartz (05-0490) [62], calcite (05-0586) [62], dolomite (11-0078), albite (10-0393) [85], and microline (19-0926) [84], which agrees with the fundamental composition of NA in Figure 2. Hatrurite (86-0402) [62], larnite (33-0302) [62] from the cement used (Figure 2), portlandite (44-1481) [62], and ettringite (37-1479) [62] from the reaction products of Ordinary Portland cement (OPC) [87,88] were also observed. Comparing the phases found using normal or carbonated water as kneading water, a sharp decrease of the phases hatrurite and larnite were observed (Inset Figure 6 labelled “C_3_S and C_2_S”, red colour “1 day normal water”, purple colour “1 day carbonated water”). Furthermore, the formation of portlandite Ca(OH)_2_ was affected by the carbonated water as kneading water (Inset Figure 7 labelled “Portlandite”, red colour “1 day normal water”, purple colour “1 day carbonated water”) and is in accordance with Equation (7). The loss of intensity of hatrurite and larnite peaks and delay in the formation of portlandite were also reported by Hou et al. [71] with acid water. The observed results can be because of the pH of the carbonated water (4.8) and decreased mechanical strength at 1 d of curing, as shown in Figure 4 and Figure 5.

Comparing the diffractogram of 1 d with those obtained at the ages of 3 and 7 d for carbonated water, the same phases were identified but an increase in the intensity was observed in the calcite phase (Inset Figure 6 labelled “CaCO_3_ (3 days)” and “CaCO_3_ (7 days)”), suggesting that carbonated water as kneading water produced carbonation [22,23,24]. This also explains the increased mechanical strength in Figure 3 and Figure 4 and DBD in Figure 5.

XRD obtained for NA using normal and carbonated water as kneading water under CC are shown in Figure 7. With carbonated water (Figure 7 inset labelled “C_3_S and C_2_S”), we observed a decrease in the peaks of the phases hatrurite and larnite. In addition, the formation of portlandite Ca(OH)_2_ was not significantly delayed when using carbonated water (Figure 8 inset labelled “Portlandite”). Both processes were due to the presence of CaCO_3_ and Ca(OH)_2_ in RMA (Figure 2). These phases acted as a buffer [5]. Hence, carbonated water can increase the mechanical properties at 1 d of age with RMA than with NA (Figure 3 and Figure 4). These results highlight that RMA, acting as a buffer for carbonated water during kneading, avoids a decrease in pH without adding CaCO_3_ or Ca(OH)_2_, as previously proposed in [5,89]. Owing to its mineralogical composition, RMA has a similar effect as CaCO_3_ and Ca(OH)_2_.

At 3 and 7 d with normal water, the same phases were identified as that of 1 d. Comparing these diffractograms with that obtained for 3 and 7 d using carbonated water, a higher intensity was observed in the calcite peaks (Figure 7 inset labelled “CaCO_3_ (3 days)” and “CaCO_3_ (7 days)”). This behaviour was already observed in the samples with NA (Figure 7) and indicates that the carbonated water produced carbonation [22,23,24]. This supports the increase in mechanical strength with carbonated water (in Figure 3 and Figure 4) and DBD (in Figure 5).

XRD obtained for NA using normal and carbonated water as kneading water under CO_2_·C are shown in Figure 8. For 1 d, with normal water, the same phases as in CC were found. For 3 and 7 d, the disappearance of the portlandite phase was observed (Figure 8 inset labelled “Effect CO_2_”), which shows the consumption portlandite when it comes into contact with CO_2_ (Equation (1)). This concurs with an increase in the mechanical strength in samples cured in CO_2_·C (Figure 3 and Figure 4). This was due to samples carbonation, as reported in [13,14,18,24,27,28,56,80]. The same phases were found for 1, 3, and 7 d with carbonated water. The portlandite also disappeared at the age of 3 and 7 d.

The effect of carbonated water at 1 d, under the CO_2_·C regime, is almost the same as that under CC (Figure 6). However, a decrease in the peaks of the hatrurite and larnite phases were observed (Figure 8 inset labelled “C_3_S and C_2_S”). For 3 d, the effect of carbonated water (Figure 6) is negligible on the calcite formed with respect to the effect produced by the carbonation chamber (Figure 8), because the amount of CO_2_ contributed by the CO_2_·C regime is greater than that of the carbonated water (Figure 8 inset labelled “CaCO_3_ (3 days)”, similar intensity found for CaCO_3_ peaks). These results agree with the delay in setting [71] and strength development [70] due to the initial decrease in pH produced by combining carbonated water and CO_2_-C regimes. At 7 d, a greater intensity was observed in the calcite phase, more with carbonated water than with normal water (Figure 8 inset labelled “CaCO_3_ (7 days)”). This indicates that pH had been regulated [22,23,24] and that carbonation of the sample is better than in with normal water and agrees with the results of the mechanical properties in Figure 4 and Figure 5 and DBD in Figure 6.

XRD obtained for RMA using normal and carbonated water as kneading water under CO_2_·C are shown in Figure 9. At 1 d, with normal water, the phases found were the same as those found in the CC regime (Figure 7). For 3 and 7 d, the disappearance of the portlandite phase was observed (Figure 9 inset labelled “Effect CO_2_”), indicating carbonation (Equation (1)) [13,14,15,16,17].

With carbonated water, the same phases were observed as that with normal water. For 1 d, a light decrease of hatrurite and larnite were observed (Figure 9 inset labelled “C_3_S and C_2_S”, red colour “1 day normal water”, purple colour “1 day carbonated water”), indicating that carbonated water has a retarding effect on the development of mechanical properties at a young age. As with NA (Figure 8), the same behaviour, including calcite peak intensities, was observed at the age of 3 d (Figure 9 inset labelled “CaCO_3_ (3)”). The low pH value of carbonated water along with accelerated carbonation (CO_2_·C) which also lowers the pH, negatively affects the strength, although less in the case of NA (Figure 3 and Figure 4). At 7 d of curing, the calcite peaks were similar with carbonated and normal water (Figure 9 inset labelled “CaCO_3_ (7)”) and agree with the mechanical properties in Figure 3 and Figure 4 and DBD in Figure 5.

### 3.5. SEM

Figure 10 shows a general SEM and elemental composition mapping of the NA mixture with normal and carbonated water as kneading water under CC at low magnification (NA-H_2_O-CC vs. NA-CO_2_·H_2_O-CC). Two main zones were detected: siliceous aggregate and cement paste. The main element in the aggregates is Si and agrees with the chemical composition (Table 4), XRD (Figure 2) results. The main elements contained in the cement paste were Ca, Al, K, and Mg. At low magnification, no differences were observed using carbonated water.

However, by increasing the magnification over the cement paste zone, significant differences were found when using carbonated water (Figure 11). First, it seems that the structure of the cement paste with carbonated water was more porous than that obtained with normal water. The qualitative analysis by SEM agrees with the highest APW found with carbonated water (Figure 5). With normal water, it can be seen that the grains with rounded faces and edges were formed around the cement particles. Nevertheless, with carbonated water, large amounts of well-developed and intertwined needles particles, with very high surface areas are observed. Considering the morphological similarities with ettringite Ca_6_[Al(OH)_6_]_2_ (SO_4_)_3_·26H_2_O, it can be speculated that the needle-like structure is a carbonated ettringite with the chemical formula Ca_6_[Al(OH)_6_]_2_ (CO)_3_·26H_2_O [37]. Because of the high CO_2_ content of carbonated water, ion exchange occurs; that is, SO_4_^2−^ is fully or partially replaced by CO_3_^2−^. A similar result was found by Pingping et al. [27] with water curing with CO_2_.

SEM images with higher magnification were taken to confirm the above results (Figure 12). With normal water, grains with rounded faces and edges were observed. However, with carbonated water, hexagonal- or orthorhombic-shaped (1) and needle-shaped particles (2) were observed. EDS analysis of the hexagonal particle revealed the presence of Ca, C, and O, indicating the possibility of CaCO_3_ [2,27]. This agrees with the greater intensity of calcite observed in XRD with carbonated water for NA (Figure 6 vs. Figure 7). For needle-shaped particles, EDS revealed a high concentration of C and O, indicating that SO_4_^2−^ was fully or partially replaced by CO_3_^2−^ to form carbonate ettringite [37]. Boumaza et al. [19] formed carbonated crystals having hexagonal or orthorhombic shapes between the needles of ettringite under a CO_2_ environment. The interlaced shape of the carbonate ettringite and greater presence of calcite (due to the carbonation produced by CO_2_ in the carbonated water) improved the mechanical properties (Figure 3 and Figure 4) compared to normal water.

Figure 13 shows a general SEM and elemental composition mapping of the RMA mixture with normal and carbonated water under CC at low magnification (RMA-H_2_O-CC vs. RMA-CO_2_·H_2_O-CC). In this case, two zones were observed: a siliceous aggregate or piece of brick, which is in accordance with the nature of the RMA (Figure 2), and cement paste containing Ca, Al, K, and Mg as the main elements. Furthermore, microcracks and a possible interfacial transition zone (ITZ), which is the area between the old and the new cement paste and is the weakest region in MRA mortar [3,90,91], were observed. These could explain the decrease in mechanical properties (Figure 3 and Figure 4) with the replacement of NA by RMA (with normal and carbonated water under CC) and the higher porosity found with RMA (Figure 5). At this magnification, no differences were found between carbonated and normal water with RMA. The same areas as with normal water are also found. This is contrary to what is observed with NA (Figure 10).

With slightly higher magnification, microcracks were more visible (Figure 14). There were fewer microcracks when using carbonated water as the carbonatation products (CaCO_3_ particles) can gradually fill pores and micropores [22,23,24,35]. This agrees with the improvement in the mechanical properties observed with carbonated water in RMA under the CC regime (RMA-H_2_O-CC vs. RMA-CO_2_·H_2_O-CC). Furthermore, this increase in carbonation products was also observed in the XRD analysis (Figure 7). Notably, when using RMA and carbonated water, the presence of carbonated ettringite was not observed, unlike when using NA (Figure 11 and Figure 12) due to the existence of calcite and portlandite in RMA (Figure 2). Calcite and portlandite act as buffers for carbonated water [5], consuming CO_2_ from carbonated water, especially portlandite (Equation (1)), thereby avoiding the full or partial replacement of SO_4_^2−^ by CO_3_^2−^.

At higher magnification (Figure 15), no microcracks were observed due to the filling of microcracks by the effect of carbonated water in the RMA. In addition, carbonate ettringite (needle-shaped particles) is not observed. The same behaviour was observed at very high magnification (Figure 16). Therefore, carbonated water on the microstructure of RMA serves the purpose of filling the microcracks. Studies on the influence of carbonated water with RMA have not been found in the literature.

### 3.6. TGA-DTA

To determine whether carbonated water produces a greater amount of CaCO_3_ in the mixes with NA and RMA in CC regime, TGA/DTA was performed (Figure 17). Five stages were observed for all the mixes with normal and carbonated water. In the stage from 480 to 1000 °C, CaCO_3_ decomposition occurred [2,56], attributed to the loss of mass resulting from calcium carbonate decomposition. A high loss of mass in this range indicates high calcium carbonate in the mix.

For the mix with NA (Figure 17A), a mass loss of 3.8% and 9.19% were observed for normal and carbonated water, respectively, (NA-H_2_O-CC vs. NA-CO_2_·H_2_O-CC) in the range of 480–1000 °C, indicating a greater amount of CaCO_3_ (product of carbonation) formation with carbonated water. This is in agreement with the mechanical properties (Figure 3 and Figure 4), DBD (Figure 6), XRD results (Figure 6), and SEM (Figure 10, Figure 11 and Figure 12). The temperature of the decomposition peak of CaCO_3_ is different between NA-H_2_O-CC and NA-CO_2_∙H_2_O-CC. This was due to the different “nature” of CaCO_3_. In the case of NA-H_2_O-CC, this CaCO_3_ is the result of the hardening process of the cement [36,92]. In the case of NA-CO_2_∙H_2_O-CC, the calcium carbonate is the result of the carbonation produced in the sample and by them exist a delayed in the decomposition temperature. In contrast, for the mix with RMA (Figure 17B), the mass loss is 5.5 and 6.01 for normal and carbonated water, respectively (RMA-H_2_O-CC vs. RMA-CO_2_·H_2_O-CC), between 480 and 1000 °C. In this case, the difference in calcium carbonate formation was not as important as in NA (although it is still greater with carbonated water than with normal water). This was already described in the analysis of the intensity for calcite peaks in XRD. The difference between the intensity of the peaks was greater in the NA mixture than in the RMA mixture, at the age of 7 d (see Figure 6 inset labelled “CaCO_3_ (7 days)” vs. Figure 7 inset labelled “CaCO_3_ (7 days)”).

## 4. Conclusions

This study presents an experimental study using carbonated water as kneading water and its impact on the physical-mechanical properties of a porous CBM made with NA and RMA. The main objective was to evaluate the influence of carbonated water together with whether or not subsequent curing in carbonation chamber on the mechanical properties and explain this behaviour using XRD, SEM, and TGA/DTA. The characterisation of the mix composition with different aggregates (NA or RMA), normal water or carbonated water as kneading water, and different hardening environments (different level of CO_2_) were performed at 1, 3, and 7 d. The following conclusions were obtained:Carbonated water worsened mechanical properties at 1 d of curing with NA under the CC regime, compared to normal water. The phases of CaCO_3_ and Ca(OH)_2_ in the RMA, acted as a buffer for carbonated water.The low pH value of carbonated water and accelerated carbonation (CO_2_·C) further lowers the pH, and negatively affects the strength at 1 d of normal curing for all the mixes. The simultaneous utilization of carbonated water as kneading water and subsequent curing in CO_2_ is not recommended.In all the mixtures studied, the effect of carbonated water increased the DBD (due to carbonation) and APW, indicating that carbonated water generated additional porosity. The carbonation reaction that occurs with carbonated water under CC explains the increase in mechanical strength at 7 d of curing for NA and RMA. A greater intensity in the CaCO_3_ peaks (XRD) and increased weight loss of calcite decomposition (TGA/DTA) was also observed.The presence of interlaced needles of ettringite carbonate observed by SEM and the increased presence of calcite (due to the carbonation produced by CO_2_ in the carbonated water) resulted in better mechanical properties than normal water. Carbonated water on the microstructure of the RMA results in the filling of microcracks (shown in the SEM images). Ettringite carbonate was not observed in this case because of portlandite in RMA, which consumed CO_2_ from carbonated water. Carbonated water as kneading water using RMA could allow for the production of precast CBM products with good mechanical properties without the need for CO_2_ curing chamber.

The utilization of carbonated water as kneading water in CBM with recycled aggregates (circular economy) can be a novel and interesting procedure to obtain a more environmentally friendly building material without the use of a carbonation chamber. At the same time, it improves mechanical properties and contributes to climate change mitigation.

## Figures and Tables

**Figure 1 materials-15-04876-f001:**
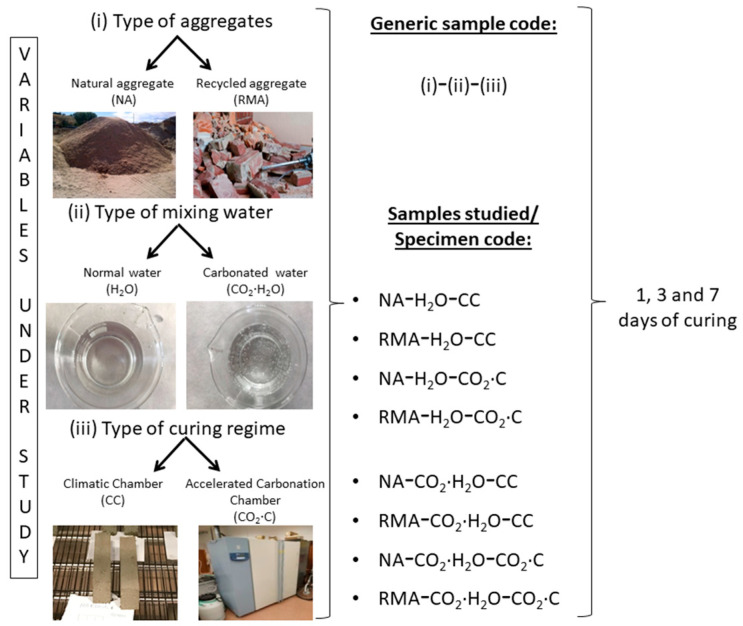
Experimental design carried out.

**Figure 2 materials-15-04876-f002:**
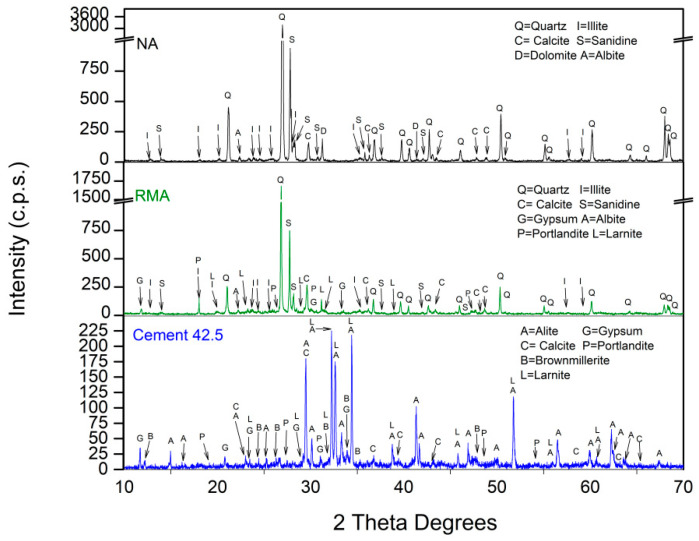
XRD patterns of NA, RMA, and cement.

**Figure 3 materials-15-04876-f003:**
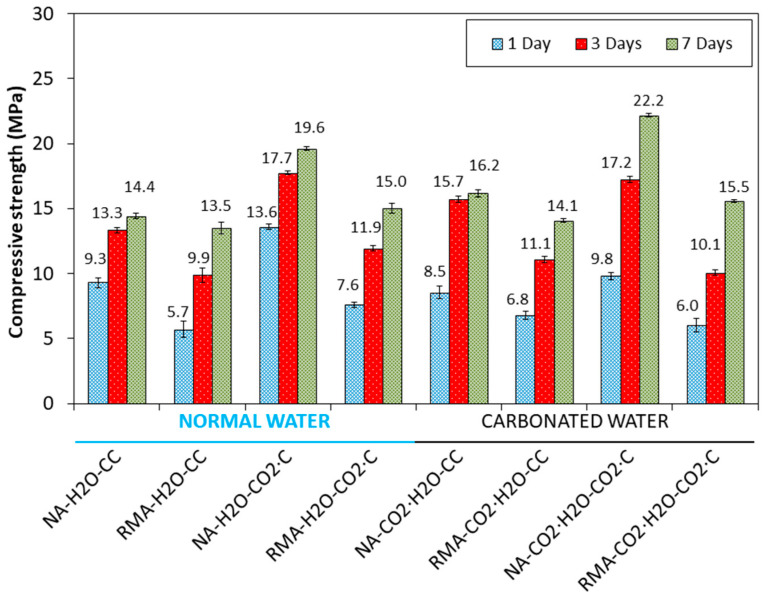
Compressive strength at different curing ages and hardening environments.

**Figure 4 materials-15-04876-f004:**
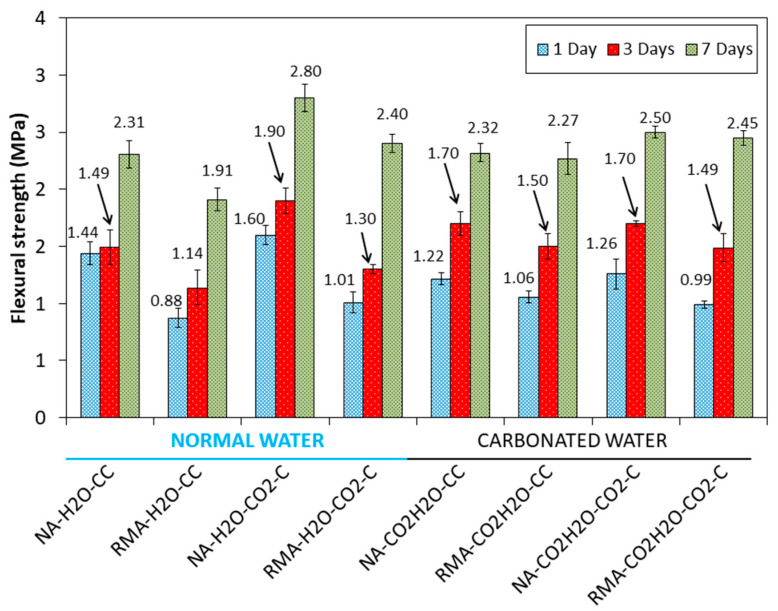
Flexural strength at different curing ages and hardening environments.

**Figure 5 materials-15-04876-f005:**
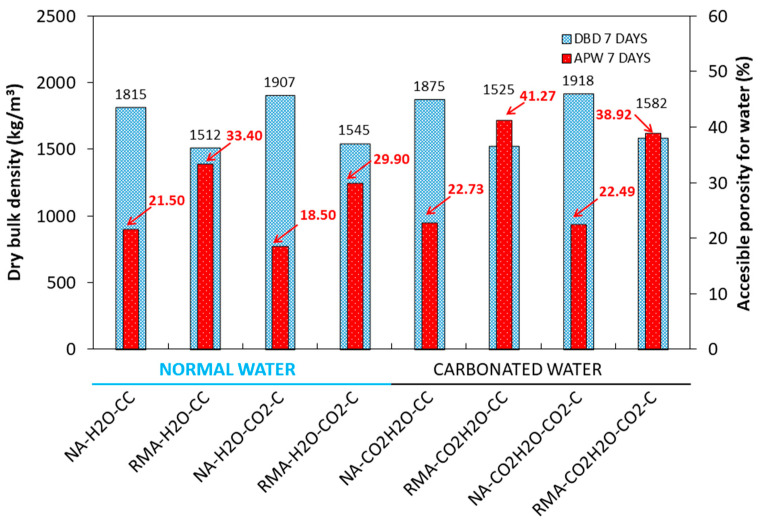
DBD and APW for 7 d under CC and CO_2_·C.

**Figure 6 materials-15-04876-f006:**
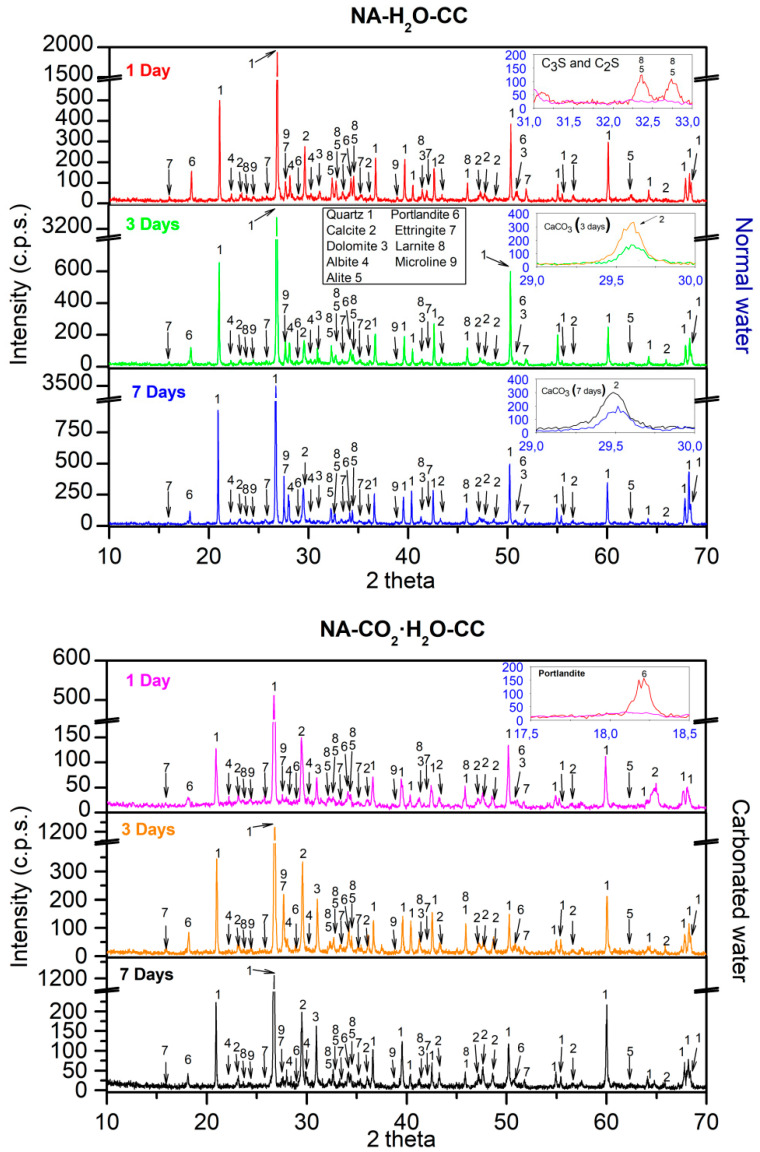
XRD for NA with normal and carbonated water under CC.

**Figure 7 materials-15-04876-f007:**
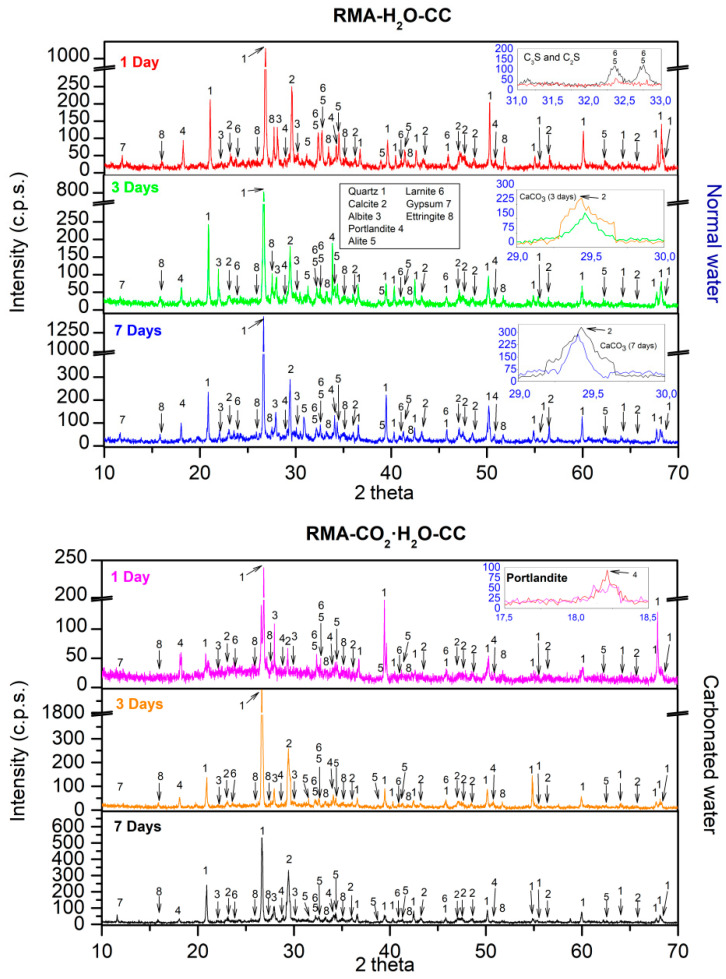
XRD for RMA with normal and carbonated water under CC.

**Figure 8 materials-15-04876-f008:**
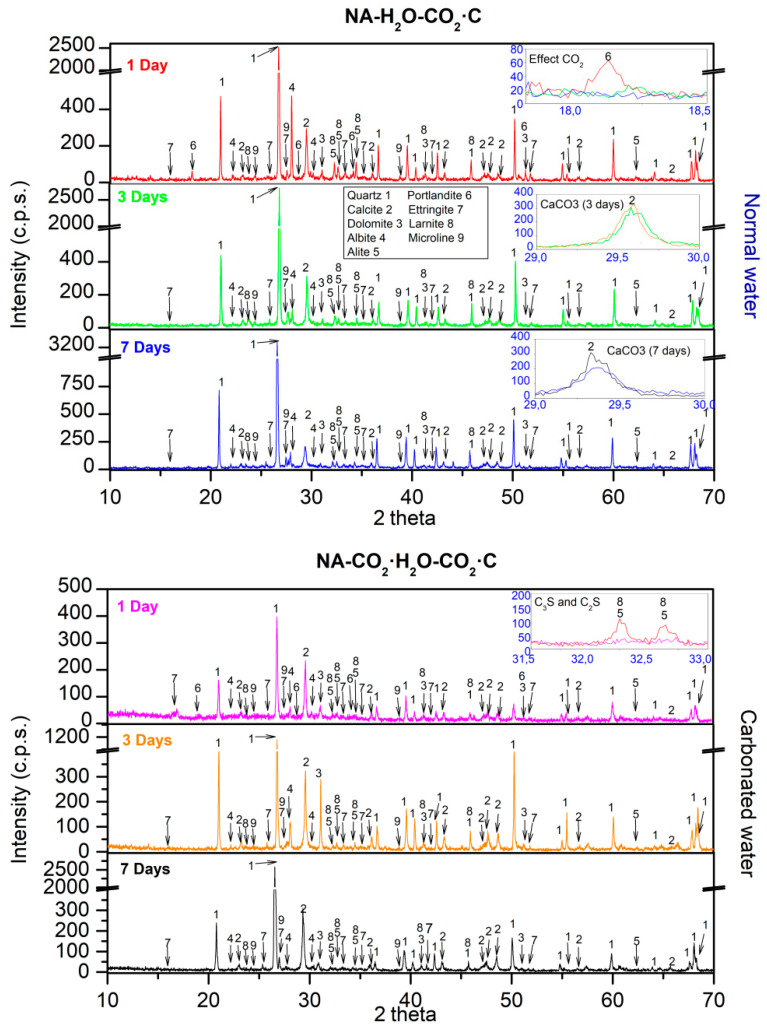
XRD for NA with normal and carbonated water under CO_2_·C.

**Figure 9 materials-15-04876-f009:**
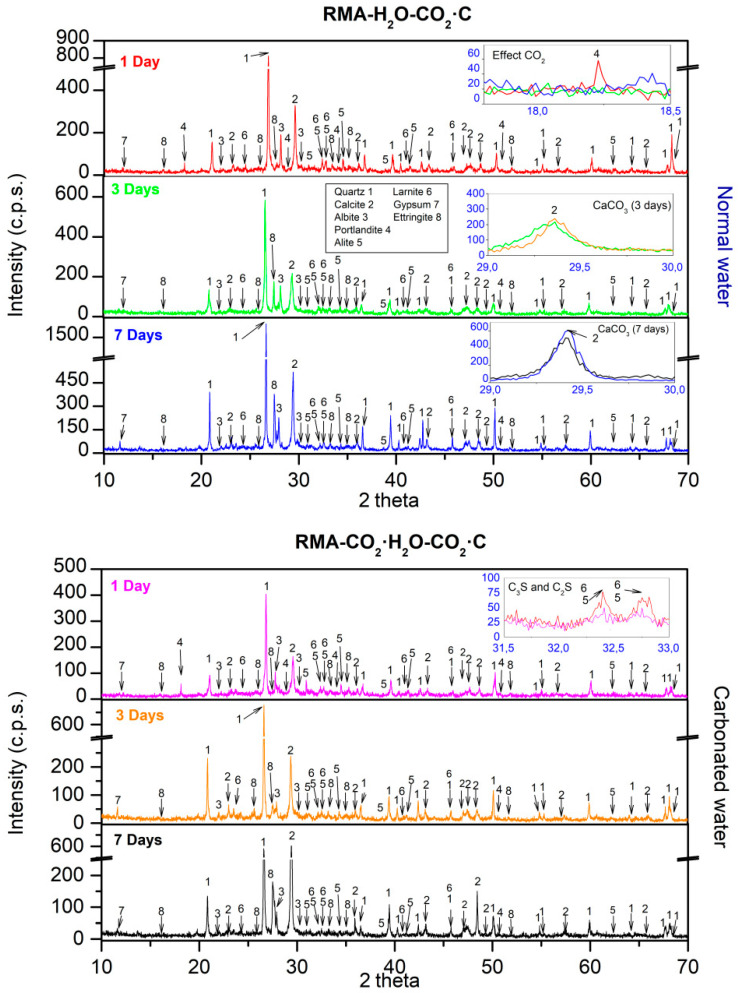
XRD for RMA with normal and carbonated water under CO_2_·C.

**Figure 10 materials-15-04876-f010:**
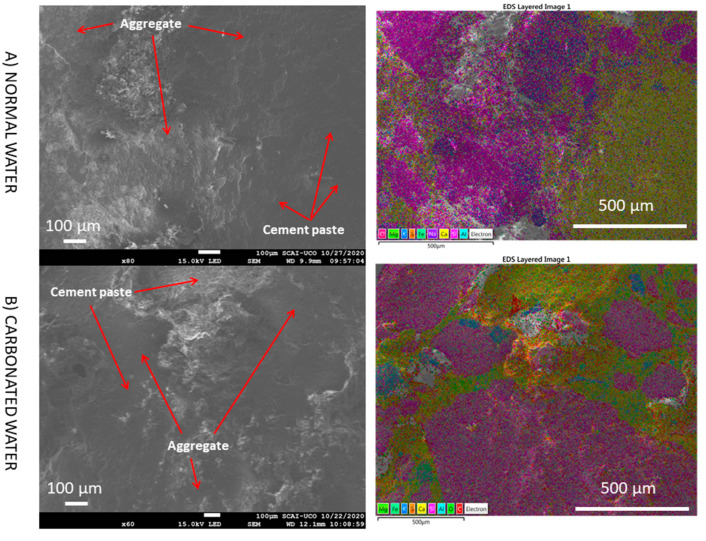
SEM images and elemental composition mapping of NA with normal and carbonated water under normal curing regime CC (NA-H_2_O-CC vs. NA-CO_2_·H_2_O-CC) at low magnification.

**Figure 11 materials-15-04876-f011:**
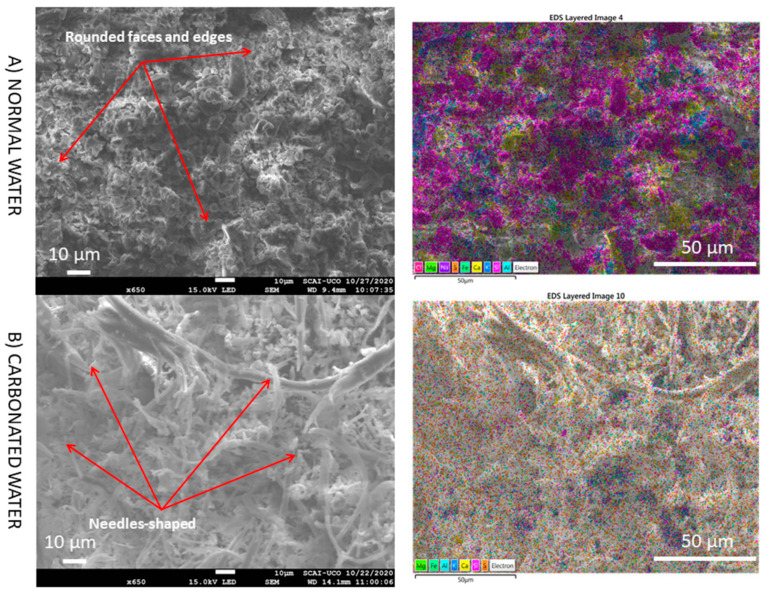
SEM images and elemental composition mapping of NA (zone cement paste) with normal and carbonated water under normal curing regime CC (NA-H_2_O-CC vs. NA-CO_2_·H_2_O-CC) at medium magnification.

**Figure 12 materials-15-04876-f012:**
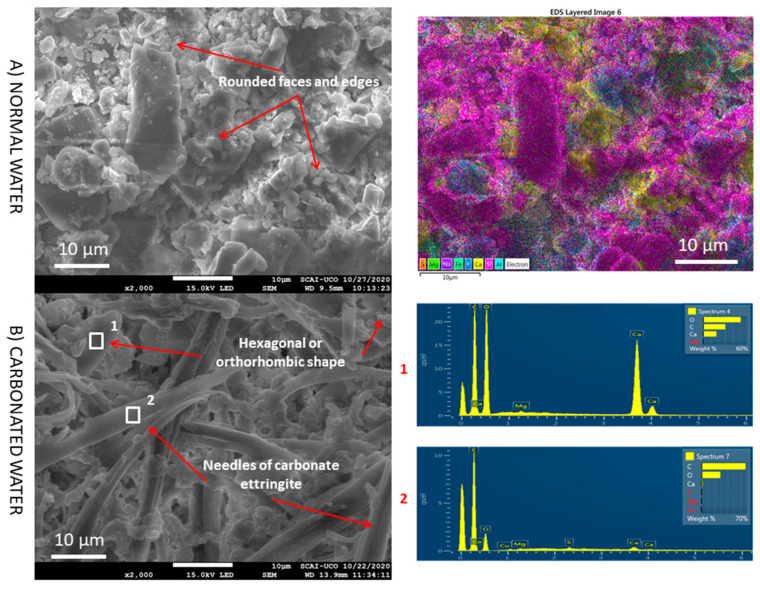
SEM images and EDS of NA (zone cement paste) with normal and carbonated water under normal curing regime CC (NA-H_2_O-CC vs. NA-CO_2_·H_2_O-CC) at high magnification. Elemental composition mapping.

**Figure 13 materials-15-04876-f013:**
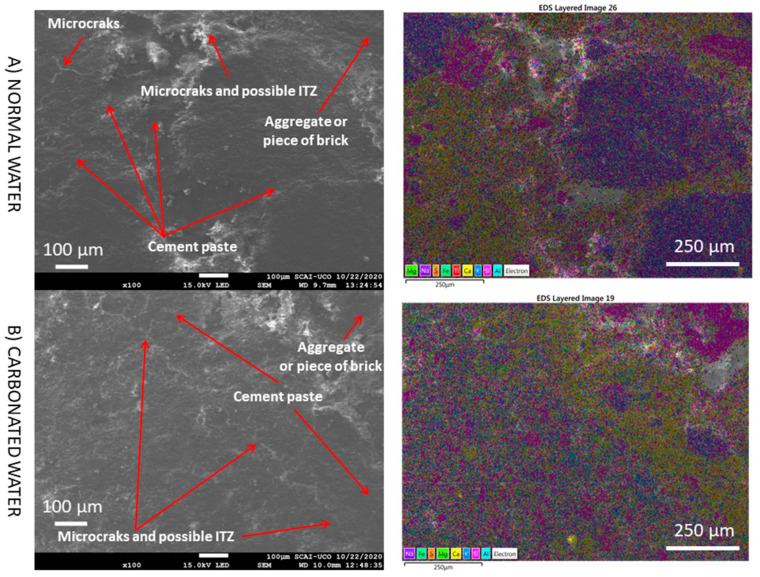
SEM images and elemental composition mapping of RMA with normal and carbonated water under normal curing regime CC (RMA-H_2_O-CC vs. RMA-CO_2_·H_2_O-CC) at low magnification.

**Figure 14 materials-15-04876-f014:**
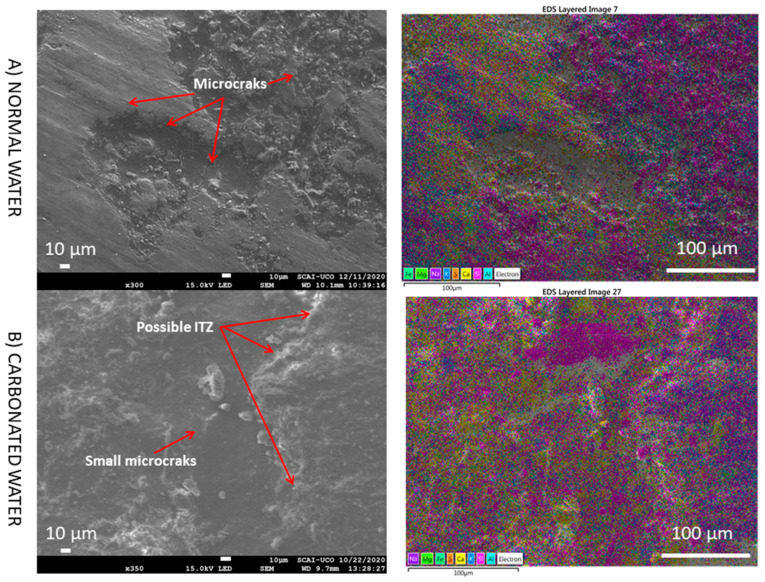
SEM images and elemental composition mapping of RMA with normal and carbonated water under normal curing regime CC (RMA-H_2_O-CC vs. RMA-CO_2_·H_2_O-CC) at medium magnification.

**Figure 15 materials-15-04876-f015:**
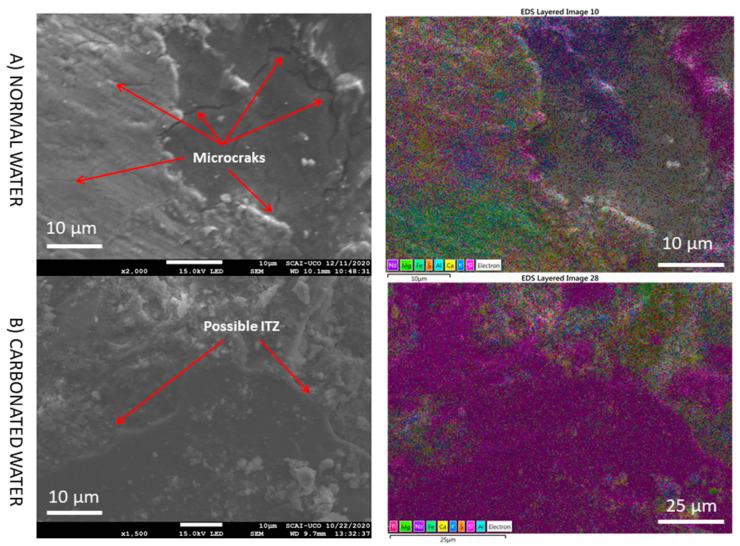
SEM images and elemental composition mapping of RMA with normal and carbonated water under normal curing regime CC (RMA-H_2_O-CC vs. RMA-CO_2_·H_2_O-CC) at high magnification.

**Figure 16 materials-15-04876-f016:**
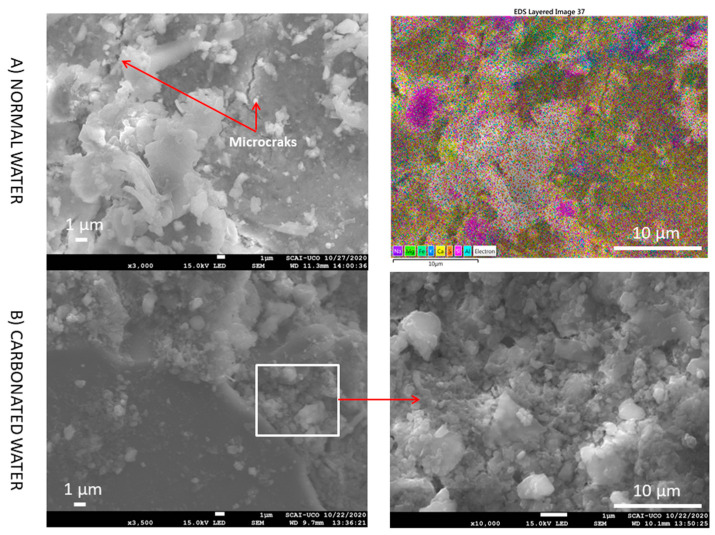
SEM images and elemental composition mapping of RMA with normal and carbonated water under normal curing regime CC (RMA-H_2_O-CC vs. RMA-CO_2_·H_2_O-CC) at very high magnification.

**Figure 17 materials-15-04876-f017:**
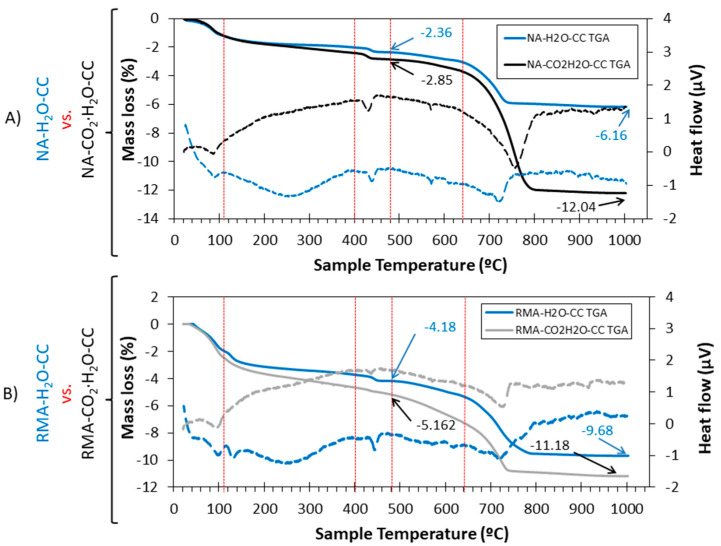
TGA (solid lines) and DTA (dotted lines) curves for (**A**) mix with NA and (**B**) mix with RMA. Use of normal and carbonated water.

**Table 1 materials-15-04876-t001:** Spindle-shaped particle size limits.

Size (mm)	ASTM C 144-04 (Limit)	Fraction Size	Original Percentage Retained	Application of 2 Gaps	Particle Size Distribution Obtained (Passing)
4	100	>4	0	0	100
2	88	2/4	12	16	84
1	62	1/2	26	35	49
0.5	32	0.5/1	30	40	9
0.25	8	**0.25/0.5**	24	0	9
0.125	1	0.125/0.25	7	9	0
0.075	0	**<0.125**	1	0	0
0.063		**TOTAL**	100	100	-

**Table 2 materials-15-04876-t002:** Aggregate weights used by different fractions.

Mixes	NA			RMA 100%	
Fraction Size (mm)	Weight Used (g)	BD * (g/cm^3^)	Volume (cm^3^)	BD * (g/cm^3^)	Weight Used (g)
>4	**0**	-	-	-	**-**
2/4	**561**	1.44	388.38	0.99	**386**
1/2	**1216**	1.49	813.53	1.05	**855**
0.5/1	**1403**	1.55	900.01	1.17	**1053**
**0.25/0.5**	**0**	-	-	-	**-**
0.125/0.25	**327**	1.38	235.80	1.19	**279**
**<0.125**	**0**	-	-	-	**-**
**TOTAL**	**3507** **(Equation (7))**				**2574**

*** BD** = **Bulk density**.

**Table 3 materials-15-04876-t003:** Weights used for the different mixes (g).

Mortar Type	NA RMA		Cement	Saturation Water		Effective Water		Total Water	w/c	Consistency Index (mm)
				H_2_O	CO_2_·H_2_O	H_2_O	CO_2_·H_2_O			
NA-H_2_O-(*)	3507	-	771	28	-	308	-	336	0.4	80 ±10
RMA-H_2_O-(*)	-	2574	771	232	-	308	-	540	0.4	80 ±10
NA-CO_2_·H_2_O-(*)	3507	-	771	-	28	-	308	336	0.4	80 ±10
RMA-CO_2_·H_2_O-(*)	-	2574	771	-	232	-	308	540	0.4	80 ±10

(*) CC or CO_2_-C.

**Table 4 materials-15-04876-t004:** XRF results for NA, RMA, and Cement.

Oxides	NA	RMA	Cement
F_2_O	-	0.74	-
Na_2_O	0.82	0.71	0.29
MgO	1.06	1.65	1.00
Al_2_O_3_	5.82	10.79	6.59
SiO_2_	49.63	34.44	18.29
P_2_O_5_	0.07	0.12	0.13
SO_3_	0.03	2.52	4.02
Cl_2_O_3_	-	0.05	0.07
K_2_O	1.52	2.18	1.09
CaO	5.60	12.18	45.61
TiO_2_	0.27	0.54	0.41
Cr_2_O_3_	0.04	0.02	-
MnO_2_	0.04	0.06	0.05
Fe_2_O_3_	1.74	3.55	2.85
CuO	-	-	0.04
ZnO	-	0.02	0.02
SrO	-	0.03	0.05
Rb_2_O	-	-	0.01
BaO	0.03	0.03	0.06
BALANCE CO_2_	32.32	30.61	19.43
TOTAL	67.68	69.39	80.57

## Data Availability

Not applicable.

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
