# Peer review of "Use of Carbonated Water as Kneading in Mortars Made with Recycled Aggregates"

_materials, 2022, doi:10.3390/ma15144876_

Round 1
Reviewer 1 Report
This research is focused in the CO2 sequestration by a carbonated water in fresh mortars with recycled aggregates. It is a very interesting and innovative research. I totally agree that it should be published after some revisions. I only have some questions/suggestions for the authors that I send in attahed.

Reviewer 2 Report
In the present paper, experimental study on the use of carbonated water as mixing water in cement mortars with the impact of re-cycled aggregates is performed. The test results are valuable for use of CO2 in production cement-based materials, such as mortar or concrete. The paper is organized well and the results are presented clearly. A number of comments on the paper are listed below.
1) Lines 150, Table 3, the unit should be occurs in the table heat.
2) Table 3, there is a type error in the last column.
3) In conclusion section, 3 or 4 note points should be provided instead of 8 points in the current form.
Reviewer 3 Report
The paper " Use of carbonated water as kneading in mortars made with recycled aggregates" is an interesting work. The paper could be improved after responding to the major comments.
2. Materials and Methods
Table 3. How to determine what is the equivalent volume of natural aggregate that occupies the recycled aggregate? Obviously, the RMA occupies more volume with less weight due to its lower density.
3. Results and discusión
It is recommended to report the amount of calcite that presents the NA and RMA
It is important to include prolonged ages (greater than 28 days), because overcarbonation attacks the C-S-H, which would compromise the mechanical properties
In this section there are no direct tests on the setting of the mixture. Include
Figures 6-9. Modify the axis of the ordinates in the same scale.
Analyze what happens with the portlandite. Is there a greater carbonation of this phase, or why is more CaCO3 generated?
Use needle-shaped crystals or needles instead of fiber.
Conclusions
In this section there are three conclusions about setting, a test that was not performed. As recommended above, it is recommended to perform this test.
Reviewer 4 Report
Dear Editor,
I apologize for the delay in sending you my revision and thank you for the opportunity of reviewing the manuscript “Use of carbonated water as kneading in mortars made with recycled aggregates”.
This study aims to present, through an experimental study, the impact on the physical-mechanical properties of porous Cement-Based Materials made with Natural Aggregate and Mixed Recycled Aggregate of carbonated water as kneading water.
More in detail
The introduction paragraph well depicts the state of the art, the environmental problems on the carbon emissions related to the construction industry and focus on the aim of this study.
The Materials and Methods paragraph appears accurate in the description of the investigated materials, the process and the analytical methods adopted. Figures and tables are helpful for the right comprehension of the experimental setting.
In my opinion, the paragraph “results and discussion” is presented in a rational way, the data, which seem to have been produced very carefully, are well presented and discussed. The pictures have good quality, the diagrams are clearly readable.
This work highlights that the use of carbonated water, such as kneading water in Cement-Based Materials with recycled aggregates, could represent an interesting procedure to obtain a more environmentally friendly building material, with the improvements in the mechanical properties of the mixes. These conclusions are well supported by appropriate evidence in this manuscript and may represent a contribution to climate change mitigation.
References are recent and relevant, correctly organized, and the appropriate key study included.
The manuscript presents interesting data that provides the necessary background information and introduces the study approach stepwise, it is well organized in proper sections, getting straight to the point and is easy to understand for all types of readers
The authors using XRD, SEM, and TGA/DTA aims to evaluate the influence of carbonated water together with whether subsequent curing in the carbonation chamber on the mechanical properties and explain this behaviour.
Reviewer 5 Report
This paper aims to evaluate the feasibility of incorporating recycled masonry aggregate (construction and demolition waste) in porous cement-based materials using carbonated water in the mixing followed by curing in a CO2 atmosphere. The content of the article is comprehensive, but some details are not well done. There are some comments below that authors may consider revising their paper.
1. This is a very interesting and creative idea, the use of carbonated water mitigates climate change. But there is no specific discussion of CO2 in this paper.
2. Line 24, “Curing in CO2 together with the use of carbonated water is not beneficial.” What does it mean? This contradicts the above.
3. Line 75, “(RCeA)” It occurs only once in the text, so no abbreviation is required.
4. Line 83, “One type of…….” There seems to be a missing sentence after this sentence? One type of…. Another type.....
5. Table 3, “NA” and “RMA” should add units
6. Table 3, what is “Saturation water” mean? An explanation should be added in Section 2.3.
7. Table 4 was omitted.
8. Line 184, The author said: " Among the causes of such behaviour may be due to released CO2 when using carbonated water that generates additional porosity and a cement paste that is less cohesive and less dense."
which should be verified by experiments such as MIP. Or add a reference to confirm.
9. Lines 186-195, When interpreting experimental results, one should not simply list the results of other studies. The results of this study should be discussed first.
For example: “the low pH value of carbonated water (4.8) can negatively affect the strength [70], delay the setting [109], and produce changes in the cement paste structure [72].” It should be explained why low pH affects early compressive strength, delay the setting, what has changed in the cement paste structure?
10. The resolution of the SEM images is too low, especially in Figures 10, 13, and 15. Is it because of the PDF production?
11. Lines 369, The porosity deduced by SEM analysis is not convincing (and should be avoided). If you want to discuss porosity, you need to access a direct porosity measurement, such as mercury intrusion porosimetry (MIP), for example. The explanation is based on qualitative and non-quantitative images, so it is not accepted as a proof.
12. Line 453, “Five stages were observed for all the mixes with normal and carbonated water.”
What are the five stages?
Only the fourth stage (calcium carbonate decomposition peak) was studied in this paper. The other four phases should also be studied in Section 3.6.
Round 2
Reviewer 1 Report
Dear authors and editor,
After the revision made by the authors I think the article should be accepted in the current form.
Best regards.
Reviewer 3 Report
The paper " Use of carbonated water as kneading in mortars made with recycled aggregates" is an interesting work. The paper could be improved after responding to the comments.
1. Introduction
- Subscript the 2 in the CO2 in the following sentence. “occurs with CO2, affects cement hydration products, and increases CaCO3 production [5,19,20]”.
2. Materials and Methods
Improve the writing “The amount of CaCO3 was calculated for NA and 111 RMA to be 197 and 239 kg/m3 respectively.”
3. Results
In table 4 shows "CO2 balance". What do you mean by CO2 balance and how was it determined?
It is important to include prolonged ages (at least 28 days), because overcarbonation attacks the C-S-H, which would compromise the mechanical properties
Reviewer 5 Report
This paper is revised。
